# Molecular Insights into Potential Contributions of Natural Polyphenols to Lung Cancer Treatment

**DOI:** 10.3390/cancers11101565

**Published:** 2019-10-15

**Authors:** Qingyu Zhou, Hua Pan, Jing Li

**Affiliations:** 1Department of Pharmaceutical Sciences, Taneja College of Pharmacy, University of South Florida, Tampa, FL 33612, USA; 2Department of Cardiovascular Sciences, Mossani College of Medicine, University of South Florida, Tampa, FL 33612, USA; huapan@health.usf.edu; 3Karmanos Cancer Institute, Wayne State University School of Medicine, Detroit, MI 48201, USA; bb8374@wayne.edu

**Keywords:** lung cancer, natural polyphenols, anticancer activities, molecular mechanisms

## Abstract

Naturally occurring polyphenols are believed to have beneficial effects in the prevention and treatment of a myriad of disorders due to their anti-inflammatory, antioxidant, antineoplastic, cytotoxic, and immunomodulatory activities documented in a large body of literature. In the era of molecular medicine and targeted therapy, there is a growing interest in characterizing the molecular mechanisms by which polyphenol compounds interact with multiple protein targets and signaling pathways that regulate key cellular processes under both normal and pathological conditions. Numerous studies suggest that natural polyphenols have chemopreventive and/or chemotherapeutic properties against different types of cancer by acting through different molecular mechanisms. The present review summarizes recent preclinical studies on the applications of bioactive polyphenols in lung cancer therapy, with an emphasis on the molecular mechanisms that underlie the therapeutic effects of major polyphenols on lung cancer. We also discuss the potential of the polyphenol-based combination therapy as an attractive therapeutic strategy against lung cancer.

## 1. Introduction

Lung cancer is the second most common cancer in both men and women and is the leading cause of cancer mortality in the United States [1], with an overall five-year survival rate of 19.4% (https://seer.cancer.gov/statfacts/html/lungb.html). The two main types of lung cancer are small cell lung cancer (SCLC) and non-small cell lung cancer (NSCLC), which account for about 15% and 85% of all lung cancer cases, respectively [2]. Tobacco smoking is the principal cause of lung cancer [3]. Secondhand smoking, chronic exposure to various occupational and environmental lung carcinogens, and previous lung diseases can also increase the risk of lung cancer [4,5,6,7,8,9]. Historically, intakes of fruits, non-starchy vegetables, whole grains, and herbs abundant in certain phytochemicals are thought to be protective against lung cancer. Epidemiological and experimental evidence suggests that natural bioactive compounds can act as chemopreventive agents to delay, suppress or reverse carcinogenic progression to advanced lung cancer through various mechanisms, including antioxidant/anti-inflammatory activities, modulation of biotransformation enzymes, anti-proliferative effect, and modulation of the immune system [10,11,12,13,14,15,16,17,18]. In addition, natural products that are derived from a variety of sources, including phytochemicals [19,20,21], hormones [22,23], and nutrients [24,25,26], have been shown to reduce the side effects and toxicities that are associated with chemotherapy and radiation therapy for lung cancer.

Although the consumption of fruits, vegetables, and natural products with a high content in anticancer natural compounds are generally considered to be beneficial in preventing and combating lung cancer, bioactive compounds from foods, or natural product extracts are not ready for uniform adoption into complementary and integrative therapy use due to inconclusive or negative results from clinical trials [27,28,29,30,31,32,33]. Over the past two decades, the rapid evolution of medical research and technologies has led to significant breakthroughs in our understanding of the molecular and genetic alterations that drive cancer development and progression. Consequently, the undergoing transition from the empirical trial-and-error medicine to precision medicine according to the unique genetic mapping of individual patients has stimulated considerable research activities in the area of natural bioactive compounds. With the advent of improved cellular and molecular experimental systems, significant progression has been made in unraveling the molecular mechanisms that underlie the antitumor properties of individual natural compounds. In this review, we focus on the compelling evidence from in vitro and in vivo studies demonstrating the ability of natural polyphenols, one of the most important groups of phytochemicals, to suppress lung cancer progression through the induction of tumor cell death and inhibition of aberrantly activated pro-proliferative and pro-survival signaling pathways. Our purpose is to highlight the potential of natural polyphenols to be developed and integrated into standard therapeutic strategies to improve clinical outcomes for patients with advanced lung cancer.

## 2. Mutations and Dysregulated Signaling Pathways in Lung Cancer

Lung cancer arises through a multistep process that is driven by the sequential accumulation of genetic mutations and epigenetic modifications, which leads to uncontrolled cell proliferation and inactivation of programmed cell death (apoptosis) [34,35,36]. As a lung tumor grows in size, the angiogenic switch is triggered so that the existing vascular network expands to form new blood vessels (angiogenesis) that are intended to sustain the nutrient and oxygen supply in the growing tumor [37,38,39]. Subsequently, lung tumor cells can acquire the ability to invade the surrounding tissues, intravasate into the circulatory system, travel to distant tissue sites, extravasate, and develop a secondary tumor at distant sites (metastasis) [40,41]. A typical lung cancer contains approximately 200 nonsynonymous mutations, some of which are considered “driver mutations” that play a dominant-acting role in tumor growth and progression [42]. For example, activating mutations in the Kirsten rat sarcoma viral oncogene homolog (KRAS) and epidermal growth factor receptor (EGFR) genes and rearrangements in the anaplastic lymphoma kinase (ALK) or ROS proto-oncogene 1 receptor tyrosine kinase (ROS1) genes are identified as oncogenic drivers in certain subtypes of NSCLC [43,44,45,46,47], so are the inactivating mutations in tumor protein p53 (TP53) and retinoblastoma 1 (RB1) genes in SCLC [48,49]. Most driver mutations are gain-of-function mutations that result in the overexpression of oncogenes or mutant proteins with dysregulated activities. For example, constitutive activation of EGFR or K-RAS that is due to mutation subsequently upregulates the Mitogen-activated protein kinase kinase (MEK)/extracellular signal-regulated kinase (ERK) and phosphoinositide 3-kinase (PI3K)/v-akt murine thymomaviral oncogene (AKT) signaling pathways, which triggers a cascade of downstream effectors promoting tumor growth, angiogenesis, and metastasis [50,51]. The p53 tumor suppressor gene is a transcription factor that activates a number of target genes to restrict aberrant cell growth through the induction of senescence, cell cycle arrest, or apoptosis [52,53]. The high expression of p53 protein is a favorable prognostic factor in a subset of patients with NSCLC [54], whereas the exon 8 mutation of p53 gene reduces the responsiveness to tyrosine kinase inhibitors (TKIs) and worsens prognosis in EGFR-mutant NSCLC patients [55]. It is well accepted that the inactivation of apoptosis plays an important role in lung carcinogenesis and resistance to treatment [56,57]. There are two main apoptotic pathways [58]. The extrinsic apoptotic pathway is initiated by the activation of multiple death receptors, such as Fas, tumor necrosis factor receptors (TNFRs), and TNF-related apoptosis-inducing ligand receptors (TRAILRs), and it proceeds through caspase-8 [59], while the intrinsic pathway is activated by mitochondrial outer membrane permeabilization (MOMP), followed by the release of cytochrome c to the cytoplasm and recruitment of pro-caspase 9 to cytochrome c [60]. The Bcl-2 family proteins, including pro-apoptotic members (such as Bad, Bak, Bax, Bcl-Xs, BID, Bik, Bim, HRK, Noxa, and PUMA) and anti-apoptotic members (such as Bcl-2, Bcl-W, Bcl-Xl, Bfl-1, and MCL-1) also regulate the intrinsic pathway [61]. The extrinsic and intrinsic pathways converge on the effector caspases (e.g., Caspases 3, 6, and 7), which are capable of cleaving hundreds of substrates, including nuclear proteins, plasma membrane proteins, and mitochondrial proteins, to trigger cell death [62]. Given that the evasion of apoptosis is one of the prominent hallmarks of cancer [63], an ideal therapeutic strategy to effectively induce apoptosis and avoid the “death by a thousand cuts” in lung cancer would be to restore p53 function and facilitate caspase activation [64,65].

## 3. Classification and Structures of Natural Polyphenols

Polyphenols are a large group of nature compounds present in plant-based foods and beverages, including fruits, vegetables, whole grains, tea, and wine. So far, more than 10,000 polyphenolic compounds have been identified [66]. Polyphenols can be classified into four main groups, including phenolic acids, flavonoids, stilbenes, and lignans, based on the number of aromatic rings, the structural elements connecting these rings to one another, and the substituents that are bound to the rings (Figure 1) [67,68]. Phenolic acids contain a single benzene rings and they can be further divided into two main subclasses, hydroxybenzoic acid and hydroxycinnamic acid derivatives that are based on the C6–C1 and C6–C3 backbones, respectively [69]. Flavonoids share a common structure with two aromatic rings (A and B) that are connected by three carbon atoms that form an oxygenated heterocycle (ring C), which is also known as the flavan nucleus or 1,3-diphenylpropane structure (C6-C3-C6 carbon skeleton) [69]. Flavonoids can be divided into six major subclasses, including flavonols, flavones, flavonones or dihydroflavones, isoflavones, anthocyanidins, flavanols, or catechins based on the variations in hydroxyl/methoxy group placements on the ring structures [68,70]. Natural stilbenes are structurally characterized by the presence of 1,2-diphenylethylene nucleus (C6–C2–C6 carbon skeleton) [71]. Lignans are a diverse group of optically active phenylpropanoid dimers, in which the two phenylpropane units are connected by the center carbon (C-8/C-8′) of their side chains [72].

The structure-activity relationships of many natural polyphenols in terms of their anticancer potential have been documented. The basic chemical structure of polyphenols contains one or more aromatic rings with one or more hydroxyl groups attached. The presence of functional phenolic hydroxyl groups makes polyphenols excellent hydrogen bond donors that confer high affinities for proteins and nuclear acids [73]. Therefore, the number and position of hydroxyl groups can have a decisive impact on the cellular bioactivities of polyphenols, which are central to their antitumoral, antimutagenic, pro-apoptotic, and antioxidant effects [74,75,76]. Considerable efforts have been devoted to the characterization of structure-activity relationships that provide the basis of rational design of polyphenol analogs with improved anticancer effect, given that the polyphenol core represents an attractive chemical structure towards new anticancer agents. For example, the cytotoxic activity is enhanced in phenols with low bond dissociation energy (BDE) values or large negative σ+ values since the inhibitory effect of simple phenols on fast-growing murine leukemia cells is related to the O–H BDE that is required to form a phenoxy radical and the brown variation of the Hammett electronic parameter (σ+) [77,78].

Structure-activity relationship (SAR), quantitative structure-activity relationship (QSAR), and docking approaches have been used to delineate the structural mechanisms that underlie the correlation between the binding affinity of polyphenol compounds for a specific oncogenic molecule and the expected anticancer activities [76]. For example, gossypol, a polyphenol that is derived from cotton seeds, is an effective inhibitor of Bcl-2, Bcl-xL, and MCL-1 with the inhibitor constant (Ki) values at sub-micromolar levels [79,80]. The elimination of the two reactive aldehydes from gossypol based on a model of the docked structure of the compound into Bcl-xL resulted in a semisynthetic analog of gossypol, namely apogossypol, which showed superior efficacy and markedly reduced toxicity in Bcl-2-transgenic mice as compared with gossypol [81,82]. A couple of studies by Wang’s group have demonstrated that gossypol forms a hydrogen bonding network with residues Arg146 and Asn143 in Bcl-2 through its aldehyde group and the adjacent hydroxyl groups on one of its naphthalene rings, while the isopropyl group on the same naphthalene ring is inserted into a hydrophobic pocket in Bcl-2 [83,84]. Based on the predicted binding model, a simplified pyrogallol-based analogue of gossypol was designed to mimic the hydrogen bonding and part of the hydrophobic interactions between gossypol and Bcl-2. Moreover, modification to the isopropyl group of pyrogallol resulted in not only improved binding affinities for Bcl-2 and Mcl-1, but also an increased cytotoxic effect on the MDA-MB-231 and PC-3 cancer cell lines with the inhibitory concentration IC50 values at sub-nanomolar levels [85].

A molecular docking study on the interaction between tea catechins and hepatocyte growth factor receptor (Met) has revealed that the gallate-containing catechins, including (−)-epicatechin gallate (ECG), (−)-epigallocatechin gallate (EGCG), and gallocatechin gallate (GCG), favorably fit into the Met binding site with hydrogen bonding being established between the aromatic hydroxyl groups of the gallate moiety and the backbone –NH of two Met kinase active sites, i.e., Met1160 and Pro1158, whereas tea catechins without the gallate group, including (−)-catechin (CAT), (−)-epicatechin (EC), and (−)-epigallocatechin (EGC), did not interact with Met1160, but exhibit affinity for the backbone –NH of Asp1222, which suggests that the gallate group is a key structure feature for binding of tea catechins to the active sites of Met kinase domain [86] The findings of the molecular docking study were confirmed by the Met kinase activity assay, which showed that ECG, EGCG, and GCG had inhibitory effects on Met kinase activity, while the other tea catechins had little or no effect [86]. Another potential molecular target of tea catechins is the proteasome, which is known to mediate the degradation of many intracellular proteins that are involved in carcinogenesis and tumor progression [87]. The SAR studies by Dou’s group showed that the ester bond-containing tea polyphenols, such as ECG, GCG, EGCG, and catechin-3-gallate (CG), were strong inhibitors for the chymotrypsin-like activity of the purified 20S proteasome with IC50 values at nanomolar levels, whereas the tea polyphenols without the gallate ester function, such as EGC, EC, gallocatechin (GC), and CAT, were unable to inhibit the proteasomal chymotrypsin-like activity [88,89].

## 4. Molecular Underpinnings of Polyphenols in Lung Cancer Treatment

The use of bioactive natural polyphenols for therapeutic prevention and intervention is an evolving strategy in the management of cancer. A thorough understanding of the mechanisms of action is imperative for integrating those polyphenols into standard oncology care. In this section, we primarily focus on the recent advances in understanding the antitumor actions of natural polyphenols in lung cancer (Figure 2). The preclinical studies were identified through a literature review that was conducted on PubMed using the key terms polyphenol and lung cancer (Table 1). Only original studies credibly investigating molecular mechanisms underlying the antitumor potential of natural polyphenols and their analogues were included.

### 4.1. Resveratrol

Resveratrol (3,5,4′-trihydroxy-*trans*-stilbene) is a naturally occurring stilbene phytoalexin that was first isolated from the white hellebore *Veratrum grandiflorum* in the 1940′s [181]. It has a wide spectrum of biological activities that confer various health-promoting effects, such as antioxidant, anti-inflammatory, antidegenerative, cardioprotective, and anticarcinogenic properties [182,183]. The anticancer activities of resveratrol are often associated with modulating enzymes that are responsible for metabolism of carcinogens, activating or inhibiting molecular targets, and signaling pathways that control cancer development and progression [184,185,186]. In lung cancer, considerable progress has been made in understanding the mechanisms by which resveratrol inhibits cell proliferation, induces apoptosis and cell cycle arrest, and suppresses invasion and metastasis (Figure 3), which highlights the potential of resveratrol to be used as a complementary treatment to augment the efficacy of existing therapies, and providing the insight into the development of novel synthetic resveratrol analogues with improved therapeutic efficacy and reduced side-effects.

The in vitro anti-proliferative effect of resveratrol is often associated with the induction of cell cycle arrest and apoptosis although the molecular underpinnings may vary among individual lung cancer cell lines. The results of the high-throughput immunoblotting (PowerBlot) and microarray gene expression profiling have revealed that the growth inhibitory effect of resveratrol was mediated by the transforming growth factor-β (TGF-β)/Smad pathway through the downregulation of the TGF-β pathway activators, Smad 2 and Smad 4, and the upregulation of the repressor Smad 7. Moreover, resveratrol-induced apoptosis and G1 phase cell cycle arrest was attributable to the activation of the caspases, the loss of mitochondrial permeability transition, and the increase in the expression of pro-apoptotic tumor suppressor p53 and cyclin-dependent kinase inhibitors p21 and p27 at both the gene and protein levels [90]. Similar results have been documented by others, which indicate that resveratrol-induced apoptosis is associated with increased expression of p53, Bax, and cleaved caspase-3 and decreased expression of Bcl-2 [91,92,93,94,95]. Resveratrol has also been shown to induce apoptosis through the downregulation of cellular FLICE (FADD-like interleukin-1 beta-converting enzyme) inhibitory protein (c-FLIP), which leads to a decrease in phospho-Akt, phospho-EGFR and NF-κB protein expression and an increase in the cleavage and upregulation of Bid, PARP, and caspase-8 and the production of hydrogen peroxide (H_2_O_2_) [96]. Besides the induction of apoptosis, resveratrol-induced premature senescence is another mechanism that is associated with its anticancer activities [97,98]. Resveratrol-induced premature senescence is correlated with increased DNA double strand breaks (DSBs) and reactive oxygen species (ROS) production in lung cancer cells [97]. When A549 and H460 cells underwent ionizing radiation and resveratrol co-treatment, resveratrol enhanced ionizing radiation-induced premature senescence and increased the ROS production in those cells [98].

The ability of resveratrol to induce apoptosis and cell cycle arrest in lung cancer cells renders it an ideal candidate for combination cancer therapy with the potential to provide additive anticancer efficacy and counteract the onset of acquired drug resistance in the treatment of lung cancer. Resveratrol has been shown to potentiate the growth inhibitory effect of cisplatin through induction of cell apoptosis, which was preceded by the depolarization of mitochondrial membrane potential, opening of the mitochondrial permeability transition pore, release of cytochrome *c*, upregulation of Bax expression, and downregulation of Bcl-2 expression [94]. Preclinical evaluation of the synergistic effect of resveratrol and the EGFR inhibitor gefitinib in a panel of human NSCLC cell lines demonstrated that resveratrol increased the sensitivity to gefitinib in all the cell lines tested, regardless of their EGFR mutation status. Moreover, resveratrol enhanced the inhibitory effect of gefitinib on EGFR phosphorylation in gefitinib-resistant PC-9 (PC9/G) human NSCLC cells by inhibiting CYP1A1 and ABCG2 protein expression, thereby increasing intracellular gefitinib accumulation [99]. Furthermore, among resveratrol and gefitinib single agent and combination treatment groups, the combination of resveratrol and gefitinib showed the highest increase in the fluorescence intensity of monodansylcadaverine (MDC), which is a marker of autophagic vacuoles, and in the number of MDC-labelled tumor cells, and in LC3B II protein expression, suggesting that the antiproliferative effect of combined resveratrol and gefitinib is, in part, attributable to the increased autophagy [99]. The molecular mechanism that underlies the synergistic effect of resveratrol and the EGFR inhibitor erlotinib appears to be different from that of resveratrol and gefitinib. Resveratrol potentiated the cytotoxic effect of erlotinib and enhanced erlotinib-induced apoptosis by repressing survivin and Mcl-1 expression, which inhibits the AKT/mTOR/S6 kinase pathway and increasing p53 and PUMA expression and caspase 3 activity [100]. Resveratrol has been shown to enhance tumor TRAIL-mediated apoptosis through a p53-independent mechanism by which resveratrol decreased the expression of phosphorylated Akt and subsequently suppressed the expression of NF-κB (p65), which leads to mitochondrial dysfunction and cytochrome c translocation [92]. When used in combination with etoposide, resveratrol counteracted etoposide-induced the upregulation of X-ray repair cross-complementing group 1 (XRCC1) expression that led to activation of Akt and ERK1/2, thereby restoring tumor cell sensitivity to etoposide [101].

Several studies have identified modulation of microRNAs (miRNAs) as one of the key mechanisms by which resveratrol exerts its antitumor activities in lung cancer [102,103,104]. Bae and coworkers identified 71 miRNAs with considerable changes in their expression levels in resveratrol-treated A549 cells while using microarray analysis [102]. Further analysis revealed that 25 of the 71 miRNAs target genes possessing experimentally confirmed function in apoptosis (97 genes), cell cycle regulation (20 genes), cell proliferation and differentiation (28 genes) [102]. Several recent studies have demonstrated the cell line-dependent functional link between the antitumor activities of resveratrol and resveratrol-regulated miRNA expression. In a study by Han et al., resveratrol treatment resulted in the upregulation of miR-622 in 16HBE-T human bronchial epithelial cells and H460 cells [103]. miR-622 was considered to be a tumor suppressor, as an increase in the expression level of miR-622 inhibited the cell proliferation and colony formation, induced cell cycle arrest at G0 phase, and delayed tumor growth in nude mice. Moreover, increase in miR-622 expression reduced K-Ras protein expression levels but had no effect on K-Ras mRNA level, suggesting miR-622 exerts its antitumor activity via targeting K-Ras [103]. In another study, Yu and coworkers examined the role of miR-520h in mediating the antitumor effect of resveratrol in CL1-4 and A549 lung cancer cells. Resveratrol was shown to induce the mesenchymal-epithelial transition (MET) by increasing the expression of protein phosphatase 2A catalytic subunit (PP2A/C) and reducing the expression of FOXC2, phospho-Akt, and p65. As the increased expression of PP2A/C was associated with downregulation of miR-520h [187], and treatment with resveratrol decreased miR-520h expression in A549 cells, it is suggested that resveratrol-induced MET and its inhibitory effect on lung cancer cell migration and invasion are attributable to its ability to inhibit miR-520h expression and activate PP2A/C, which in turn suppresses Akt-mediated activation of the NF-κB pathway, which promotes the malignant behaviors of lung cancer cells [104].

Based on the identified structure-activity relationship, analogues of resveratrol have been synthesized and tested for their antitumor activities in lung cancer cell lines. A synthetic resveratrol named BCS (3,4,5-trimethoxy-4V-bromo-*cis*-stilbene), in which the hydroxyl group of resveratrol is substituted by the methoxy group, was about 1100 times more potent than resveratrol in the growth inhibition of A549 cells (IC_50_; 0.03 µM vs. 33 µM) [105]. The anti-proliferative effect of BCS was highly associated with cell cycle arrest at G2/M phase and the induction of apoptosis possibly through a mitochondrial-mediated pathway, as manifested by the elevated expression levels of p53 and p21, and the release of cytochrome c in the cytosol [105]. SS28 ((E)-1,2,3-trimethoxy-5-(4-methylstyryl)benzene (6 h)) is a resveratrol-based tubulin inhibitor that exerts its antiproliferative activity by binding to its cellular target tubulin to disrupt the microtubule dynamics [120]. SS28 treatment induces G2/M cell cycle arrest by inhibiting tubulin polymerization during cell division and it leads to apoptosis via the intrinsic (mitochondrial) pathway, as indicated by the loss of mitochondrial membrane potential and activation of Caspase 9 and Caspase 3 [120]. Another resveratrol analogue, 4,4′-dihydroxy-*trans*-stilbene (DHS), significantly suppressed tumor growth and angiogenesis in C57BL/6 mice bearing Lewis lung carcinoma (LLC) and inhibited the anchorage-dependent or -independent LLC cell growth in both mouse and zebrafish lung cancer invasion models [121]. In addition, the results of the in vitro study showed that DHS inhibited LLC cell proliferation, migration, and invasion, and induced the accumulation of hypodiploid cells in the sub-G1 phase, which suggests that the antitumor effect of DHS is via inhibiting DNA synthesis and driving cells towards the apoptotic pathway [121].

### 4.2. Tea Catechins

Catechins belong to a family of flavonoids and they are the main component of green tea in which catechins comprise 80–90% of the flavonoids, with EGCG being the most abundant catechin (up to 60%) and EGC being the second most abundant (up to 20%), followed by ECG (up to 14%) and EC (about 6%) [188,189]. With a structure of two benzene rings (the A- and B-rings) and a dihydropyran heterocycle (the C-ring) with a hydroxyl or galloyl group over carbon 3, catechins have four possible diasteroisomers. Two isomers with *trans* configuration are called catechin ((+)-catechin and (−)-catechin), and two with *cis* configuration, called epicatechin ((+)-epicatechin and (−)-epicatechin) [190]. The number and configuration of hydroxyl groups on the B ring are the most important determinants of the antioxidant ability of catechins, while the presence of the galloyl group might further increase the antioxidant action [191,192,193].

The mechanism underlying the inhibitory effects of tea catechins, especially EGCG, on lung cancer progression have been extensively investigated [194,195,196,197]. In general, the growth inhibition effect of EGCG is superior to EGC, ECG, and EC [125], and it is associated with the induction of G_2_-M arrest [126] and activation of p53-dependent transcription [125]. EGCG treatment has been shown to effectively inhibit the in vitro and in vivo growth of fusion gene- or EGFR-driven lung cancer cells such as H2228 and HCC78 cells that harbor the EML4-ALK fusion gene and SLC34A2-ROS1 fusion gene, respectively, and PC-9, RPC-9, and H1975 cells that harbor EGFR^19DEL^, EGFR^19DEL + T790M^, and EGFR^L858R + T790M^ mutations, respectively [127], which suggests that EGCG has a broad growth inhibitory effect independent of the EGFR mutation status and the ALK or ROS1 fusion status. Although the results of the in vitro study showed that the anti-proliferative activity of EGCG was attributable to the suppressed phosphorylation of EGFR, ALK, and ROS1, and their downstream proteins, Akt and ERK, the in vivo growth inhibitory effect of EGCG in xenograft tumors was associated with the inhibition of HIF-1α expression and reduction tumor angiogenesis, which suggests that tumor response to EGCG is influenced by the tumor microenvironment [127].

The results from a human cancer cDNA expression array study showed that EGCG downregulated the expression of 12 genes and upregulated the expression of four genes out of the 163 genes examined [128]. Among the 12 downregulated genes, two genes (NF-κB inducing kinase (NIK) and death-associated protein kinase 1 (DAPK1)) are associated with apoptosis, two genes (MAP kinase p38γ and CDC 25B/M-phase inducer phosphatase 2) associated with cell cycle, two genes (envoplakin and synapse-associated protein 102 (SAP102)) related to cell-cell interaction, three genes (Rho B, T-lymphoma invasion and metastasis inducing protein 1 (TIAM1) and Cdc42 GTPase-activating protein (Cdc42GAP)) related to the Rho family of small GTPase and regulator, tyrosine–protein kinase (SKY) gene, dishevelled 1 gene, and *EGFR* gene [128]. The four EGCG-upregulated genes included retinoblastoma binding protein (RBQ1), *VEGF*, retinoic acid receptor α1 (RAR-α1), and insulin-like growth factor-binding protein 3 (IGFBP 3) genes [128]. It is noteworthy that high levels of IGFBP 3 in plasma are associated with reduced lung cancer risk [198].

A line of evidence has shown changes in miRNA expression in response to EGCG treatment in lung tumor cell lines and 4-(methylnitrosamino)-1-(3-pyridyl)-1-butanone (NNK)-induced mouse lung tumors [129,199]. EGCG treatment resulted in increased miR-210 expression, leading to reduced proliferation and anchorage-independent growth in CL13 mouse lung adenocarcinoma cells and H460 and H1299 human NSCLC cells [199]. Moreover, EGCG increased the activity of both mouse and human miR-210 gene promoters in H1299 and H460 cells that were transfected with the 2 kb of mouse and 600 bp of human miR-210 gene promoter driven luciferase reporters [199]. Furthermore, the activity of HRE-luciferase in response to EGCG was increased from 2000–4500 U to 8000–40,000 U with the addition of HIF-1a expression vector, and an increased HIF-1α protein expression was observed in EGCG-treated lung cancer cells, which suggests that the upregulation of miR-210 by EGCG is mediated through the HRE in the promoter of miR-210 and stabilization of HIF1α [199]. The involvement of miRNA-mediated gene regulation in antitumor activities of EGCG has also been demonstrated to modulate miRNA expression in 4-(methylnitrosamino)-1-(3-pyridyl)-1-butanone (NNK)-induced mouse lung tumor model [200]. The results of the miRNA microarray showed that 12 miRNAs were upregulated in response to the EGCG treatment, while 9 miRNAs were downregulated. ECGC treatment was found to induce changes in the expression of 21 mRNAs in NNK-induced mouse lung tumors. Moreover, a group of 26 genes were identified as potential targets of the EGCG-regulated miRNAs. Changes in the expression levels of those genes were inversely correlated to changes in the expression levels of the corresponding miRNAs [200]. Further analysis of the role of the 26 miRNA targeted genes revealed an interaction network that is centralized by IGFBP5 and is involved in the regulation of Akt, MAP kinases, NF-κB, and cell cycle [200], which is consistent with the documented mechanisms of inhibitory effects of EGCG on cell cycle and inflammation [194,201]. It was noted that EGCG-induced the upregulation of miR-210 in cultured lung tumor cells was not one of the 26 miRNAs which expression levels were significantly altered in response to EGCG treatment in vivo. This discrepancy could be due to differences in the oxidative stress levels and EGCG-binding proteins between in vitro cultured cells and primary tumors, and in EGCG bioavailability and the elimination half-life between the in vitro and in vivo systems [200].

EGCG has been proved to be beneficial in combination with cancer preventive and chemotherapeutic agents [130,131]. Cotreatment of EGCG with celecoxib, a cyclooxygenase-2 selective inhibitor, synergistically induced apoptosis through the upregulation of *GADD153* gene expression and activation of the mitogen-activated protein kinase (MAPK) signaling pathway [130]. EGCG in combination with cisplatin significantly inhibited cell proliferation and induced cell cycle arrest in G1 phase and apoptosis in cisplatin-resistant A549 cells and suppressed the growth of cisplatin-resistant A549 xenograft tumors [131]. The mechanism of resensitization of tumor cells to cisplatin by EGCG is linked to the inhibition of DNA methyltransferase (DNMT) activity and histone deacetylase (HDAC) activity, reversal of hypermethylated status, and downregulated expression of the GAS1, TIMP4, ICAM1, and WISP2 gene [131]. The combined treatment of EGCG and another dietary polyphenol, luteolin, resulted in synergistic/additive apoptotic and growth inhibitory effects in both in vitro and in vivo lung tumor models. It was noted that p53 wildtype lung cancer cell lines showed greater sensitivity to co-treatment with EGCG and luteolin than p53-mutant or p53-null cell lines, and the combination effectively increased stabilization and ATM-dependent S15 phosphorylation of p53 and mitochondrial translocation of p53. Those results suggest that p53 is required for apoptosis that is induced by the combination of EGCG and luteolin [132].

Similar to EGCG, green tea extract has been demonstrated to exert anticancer activities across a spectrum of lung cancer cell lines and in vivo tumor models through different mechanisms, including the induction of apoptosis through upregulation of p53 expression and downregulation of Bcl-2 expression [133,134,135,136,137], and the inhibition of tumorigenesis through inhibition of cyclooxygenase-2, inactivation of Akt and NF-κB and degradation of IκBα [138], and through the induction of dominant-negative activator protein 1 (*TAM67*) and inhibition of activator protein-1 (AP-1) pathway [139]. The results of the proteomic analysis of A549 cells treated with green tea exact reveals 14 proteins with a ≥2-fold change in the expression level are involved in calcium-binding, cytoskeleton and motility, metabolism, detoxification, or gene regulation [140]. In particular, green tea extract was found to upregulate the expression of lamin A/C, which regulates actin polymerization in nucleus, which leads to decreased cell motility and growth and increased apoptosis [140]. Combination of polyphenon E (a standardized green tea polyphenol preparation) and atorvastatin, an inhibitor of 3-hydroxy-3-methylglutaryl CoA reductase that is commonly used for the treatment of hypercholesterolemia, synergistically inhibited 4-(methylnitrosaminao)-1-(3-pyridyl)-1-butanone induced lung tumorigenesis in mice and the tumor cell proliferation through enhanced apoptosis, which implicates that the combined use of green tea polyphenols and atorvastatin might be beneficial in lung cancer prevention and therapy [141].

### 4.3. Curcumin

Curcumin is a bioactive phytochemical in the dietary spice turmeric and it has potential anticancer activity against various types of cancer, including lung cancer [202,203]. The results from the in vitro studies have demonstrated that curcumin treatment inhibits tumor cell growth by inducing apoptosis through a variety of p53-independent and mitochondria-dependent pathways [143,144,145,146,147]. In a study by Wu et al., curcumin treatment in cultured H460 cells resulted in cell cycle arrest at the G2/M phase, initial upregulation, followed by the downregulation of cell cycle regulator cyclin D and E, upregulation of Bax, Bad and FAS/CD95 and downregulation of Bcl-2, Bcl-xL, and XIAP protein expression, increase in ROS, intracellular Ca^2+^ and endoplasmic reticulum stress, which led to a loss of mitochondrial membrane potential (ΔΨ_m_) and activation of caspase-3, release of growth arrest and DNA damage inducible gene 153 (*GADD153*) and glucose-regulated protein 78 (*GRP78*) from mitochondria to cytosol and nuclei, and decreased CDK1, CDK2, CDK4, and CDK6 protein expression and increased caspase 8 and *Endo G* mRNA expression [145]. Similar results elucidating the mechanism underlying the apoptotic activity of curcumin have been reported by other research groups using different tumor cell lines. For example, curcumin inhibited the proliferation of A549 cells through upregulation of Bax and downregulation of Bcl-2 expression, and activation of the mitochondrial apoptosis pathway, as manifested by the decreased mitochondrial membrane potential and increased release of cytochrome C from mitochondria to cytoplasm [146]. The effect of curcumin on inhibiting cell growth and inducing cell cycle arrest at the G1/S phase and apoptosis in PC-9 cells has been associated with the upregulation of the expression of *GADD45*, *GADD153*, CDK inhibitors *p21* and *p27* genes, and downregulation of the expression of *cyclin D1*, *CDK2*, *CDK4*, and *CDK6* genes [144]. A501, a synthetic analogue of curcumin with improved anticancer activities, induced cell cycle arrest at the G2/M phase by decreasing the expression of cyclin B1 and cdc-2, and promoted apoptosis by increasing the expression of p53 and Bax and decreasing the expression of Bcl-2 [153].

Although apoptosis induction appears to be the main mechanism underlying the antitumor activities of curcumin in lung cancer, there is evidence of other mechanisms being involved in the inhibitory effect of curcumin on lung tumor survival and progression. The anti-proliferative effect of curcumin has been associated with the inactivation of the PI3K/Akt/mTOR signaling pathway [148,149], upregulation of miR-192-5p [148], and induction of autophagy [149,150]. In a study by Liao et al., the ability of curcumin to suppress the proliferation, invasion, and metastasis of A549 cells was attributable to its inhibitory effect on the expression of GLUT1, MT1-MMP, and MMP2 in A549 cells [151]. Targeting GLUT1 has been sought as an attractive approach for cancer therapy, as upregulation of GLUT1 expression in malignant tumor cells is known to be responsible for the increased glucose uptake needed to drive ATP production through aerobic glycolysis, also known as the “Warburg effect” [204,205,206]. However, the overexpression of GLUT1 in A549 cells was found to attenuate the inhibitory effect of curcumin against tumor cell invasion in vitro and metastasis in vivo and increase the intracellular expression levels of MT1-MMP and MMP2, implicating that curcumin inhibits lung tumor growth and metastasis through its modulatory effect on the GLUT1/MT1-MMP/MMP2 pathway, but not by targeting GLUT1. In addition, it is suggested that GLUT1 overexpression might potentially confer resistance to curcumin treatment in lung cancer [151]. In another study by Tsai et al., the anti-migratory and anti-invasive effect of curcumin was attributable to the inhibited adiponectin expression via blockage of the adiponectin receptor 1 expression, the inactivated p38 and ERK pathways, and the downregulated expression levels of p65, MMP-2, -9, -3, -13, and -14 [152]. Given the additional evidence indicating that adiponectin regulated NF-κB expression through the Akt pathway, it was concluded that curcumin inhibited lung cancer metastasis through the adiponectin/NF-κB/MMP signaling pathway [152].

The potential of curcumin in combination with other anticancer drugs for lung cancer treatment has been documented. Curcumin potentiated the anti-proliferative effect of gefitinib in three gefitinib-resistant NSCLC cell lines, including CL1-5 (EGFR^wt^), A549 (EGFR^wt^), and H1975 (EGFR^L858R + T790M^) cell lines through blockage of EGFR activation and induction of EGFR degradation [154]. Moreover, curcumin enhanced the antitumor effect of gefitinib in CL1-5, A549, and H1975 xenografts in vivo. Notably, curcumin alone, and in combination with gefitinib, decreased the protein expression of EGFR and Akt in CL1-5 xenografts, which was not affected by gefitinib treatment alone. In addition, co-treatment with curcumin reduced the gefitinib-induced villi damage and apoptosis in mouse intestines possibly through the modulatory effect of curcumin on gefitinib-induced p38 activation [154]. Combined treatment with curcumin and carboplatin resulted in synergistic effect on cell proliferation, apoptosis, invasion, and migration [155]. This synergism appeared to be mediated by multiple mechanisms, including efficient downregulation of MMP-2 and MMP-9, substantial suppression of NF-kB via the inhibition of the Akt/IKKa pathway and enhanced ERK1/2 activity, augmented apoptosis induction through increased upregulation of p53 and p21, and downregulation of Bcl-2 protein expression [155].

### 4.4. Quercetin

Quercetin, a plant pigment and the most abundant dietary flavonol, is known to possess anti-proliferative and proapoptotic effects against many human cancers, including lung cancer [207]. It has been demonstrated that quercetin induces cytotoxicity and apoptosis in human NSCLC cells through multiple mechanisms. The mechanisms by which quercetin induces cell growth inhibition, cell cycle arrest at the G2/M phase and apoptosis involve the increase in the expression levels of survivin, cyclin B1, phospho-cdc2 (threonine 161), total p53 (DO-1), phospho-p53 (serine 15) and p21 proteins, and the induction of abnormal chromosome segregation [156]. Besides inactivation of Akt, quercetin-induced cleavage of caspase-3, caspase-7 and PARP has been found to be accompanied by the increased phosphorylation of MEK, ERK, c-Jun, and JNK, which suggests that the activation of the MEK-ERK pathway plays an important role in quercetin-induced apoptosis [157]. The results of the microarray analysis of quercetin-regulated genes in H460 cells revealed that quercetin upregulated genes that are associated with cell cycle arrest (p21^Cip1^, GADD45), the death pathway (including TRAILR, FAS, TNFR1), the JNK pathway (MEKK1, MKK4, JNK), the IL1 receptor pathway (IL1, IL1R, IRAK), the caspase cascade (caspase-10, DFF45), and the NF-*κ*B pathway (I*κ*B*α*), while it downregulated genes that are involved in cell survival (NF-*κ*B, IKK, AKT) and proliferation (SCF, SKP2, CDKs, cyclins) [158].

Quercetin has been demonstrated to exert anti-invasive and anti-metastatic activities in lung tumor cells through the downregulation of monocarboxylate transporter 1 (MCT1) [159], inhibition of aurora B kinase activity and histone 3 phosphorylation [160], disassembly of microfilaments, microtubules, and vimentin filaments along with the inhibition of vimentin and N-cadherin expression [161]. Quercetin treatment effectively suppressed the in vitro migration/invasion and in vivo bone metastasis of NSCLC cells by increasing the expression of the epithelial marker, E-cadherin, and decreasing the expressions of the mesenchymal markers, N-cadherin, and vimentin [162]. The mechanism that is associated with quercetin inhibited cell motility involved F-actin-containing microfilament bundle rearrangement and the suppression of EMT through both Snail-dependent Akt activation and Snail-independent ADAM9 pathway [162]. A recent study on the effect of five phytochemicals, including quercetin, curcumin, chrysin, apigenin, and luteolin on NiCl_2_ (Ni)-induced the migration and invasion of cultured lung cancer cells revealed that the most efficient phytochemical compound inhibiting cell migration and invasion was quercetin, followed by chrysin and apigenin [163]. Further investigation demonstrated that quercetin and chrysin at 2 and 5 μM significantly suppressed Ni-induced rise in Toll-like receptor 4 (TLR4) expression, nuclear p65 level, and relative phospho-IKK-β and phospho-IKK-α levels, which suggests that the anti-invasive effect of quercetin is associated with the downregulation of TLR4/NF-κB signaling pathway [163].

Quercetin treatment in combination with Trichostatin A, a histone deacetylase inhibitor, significantly increased growth arrest and apoptosis through the mitochondrial pathway in A549 cells expressing wild-type p53, but not in H1299 cells harboring a p53 null mutation [164]. Moreover, quercetin treatment enhances TSA-induced acetylation of histones H3 and H4 through the p53-independent mechanism [164]. Cotreatment of quercetin with gemcitabine, a pyrimidine nucleoside analogue that inhibits DNA synthesis, promoted apoptosis via the inhibition of heat shock protein 70 (HSP70) expression [165]. It is evident that quercetin-induced HSP70 inhibition is associated with the caspase-dependent apoptosis through intrinsic apoptotic pathway, given the fact that quercetin-induced HSP70 inhibition significantly increased the caspase-3 activity, while the combination of quercetin and gemcitabine significantly increased caspase-9 activity [165]. HSP70 is known to control proteostasis and anti-stress responses in rapidly proliferating tumor cells and thus reduce the sensitivity of tumors to conventional anti-cancer drugs [208]. The mild toxicity profile of quercetin and its potential to act as a HSP70 inhibitor render it an attractive agent for use as part of a combination regimen to improve tumor response to chemotherapy with less severe side effects.

### 4.5. Other Naturally Occurring Polyphenols

Thymoquinone (TQ), the predominant bioactive constituent that is present in black seed oil (Nigella sativa), and Caffeic acid phenethyl ester (CAPE), a phenolic compound that is isolated from propolis, have been shown to induce G2/M cell cycle arrest and apoptosis through mechanisms similar to those of resveratrol [95]. Notably, all three agents (i.e., CAPE, TQ, and resveratrol) decreased the expression of cyclin D and increased the expression of TRAIL receptor 1 and 2, and p21 with the highest increase in p21 expression being observed in TQ-treated A549 cells. Moreover, CAPE and TQ upregulated Bax expression, while TQ and resveratrol downregulated Bcl-2, NF-κB, and IKK1 expression in A549 cells [95]. Based on those findings, further studies are warranted to evaluate the potential benefit of using TQ and CAPE in combination with other therapeutic agents for the treatment of lung cancer.

Pterostilbene (*trans*-3,5-dimethoxy-4′-hydroxystilbene), a naturally derived phytoalexin and a demethylated analog of resveratrol, was shown to inhibit A549 cell proliferation and induce S-phase cell cycle arrest by activating the ATM/ATR-CHK1/2-p53 signaling pathway [166]. Moreover, in two precancerous human bronchial epithelial cell lines, HBECR and HBECR/p53i, which have normal and reduced p53 expression levels, respectively, low-dose pterostilbene (at 1 and 5 μM) inhibited cell growth and induced cell cycle arrest in S phase and senescence in HBECR cells more efficiently than in HBECR/p53i cells, which suggests that the chemopreventive activity of pterostilbene is p53-dependent. This finding implicates that the use of pterostilbene as a chemopreventive agent for squamous lung carcinogenesis should be initiated at the early stage before p53 mutation occurs. Another analogue of resveratrol, bakuchiol (1-(4-hydroxyphenyl)-3,7-dimethyl-3-vinyl-1,6-octadiene) that was isolated from the seeds of *Psoralea corylifolia* L. (Leguminosae), exhibits a more significant cytotoxic effect in A549 cell line than in EA.hy926 endothelial cells, HUVECs, and primary cultured mouse embryo fibroblasts. It induces apoptosis and cell cycle arrest in S phase by increasing ROS production, interrupting mitochondrial homeostasis, increasing Bax/Bcl-2 ratio, upregulating p53, and activating Caspase 9/3, which suggests that the apoptotic effect of bakuchiol is p53-dependent and involves a mitochondrial-mediated pathway [167].

Chlorogenic acid (CGA) is the ester of caffeic acid and (−)-quinic acid, one of the most abundant phenolic acid compounds found in coffee and tea [209]. A substantial body of evidence has indicated that CGA exerts antioxidant [210,211], anti-inflammatory [212], antidiabetic [209], antimicrobial [213,214], and anticancer [215,216] activities. Different mechanisms that are associated with the antitumor properties of CGA have been proposed, including enhancing the activity of aryl hydrocarbon hydroxylase, suppressing the oxidative formation of 8-hydroxy-2’-deoxyguanosine (8-OH-dG) in DNA, reducing the production of ROS, and regulating the immune system [217,218,219]. A study by Part et al. demonstrated that CGA significantly decreased the HIF-1α protein level without changing its mRNA level in A549 cells under hypoxic conditions and subsequently suppressed the transcriptional activity of HIF-1α, leading to decreased expression of its downstream target VEGF [168]. Moreover, CGA inhibited hypoxia-stimulated HUVEC migration, invasion, and tumor formation in vitro and VEGF-stimulated angiogenesis in Matrigel plugs in vivo through the mechanism of inhibiting the HIF-1α/AKT signaling pathway [168]. The observed antiangiogenic potential of CGA suggests that CGA could be a novel therapeutic option for the treatment of lung cancer.

Fisetin (3,3′,4′,7-tetrahydroxyflavone), a naturally occurring diet-based flavonoid, exerts anticancer activity against different cancer cell lines, including NSCLC cell lines, when used alone or in combination with other chemotherapeutic agents [169,170,171,220,221,222,223]. The inhibitory effect of fisetin on lung tumor cell growth is attributable to dual suppression of PI3K/Akt and mTOR signaling, as evidenced by the activation of PTEN, phospho-AMPKα, and TSC2, and the inhibition of PI3K, phospho-Akt, phospho-mTOR, and several downstream targets of mTOR [169]. Fisetin was shown not only to inhibit the growth and induce the apoptosis of A549 cells with acquired cisplatin resistant, but also enhance the cisplatin cytotoxicity in cisplatin-resistant cells through the modulation of the MAPK/survivin/caspase pathway [170]. Fisetin showed a synergistic effect with paclitaxel on growth inhibition and mitotic catastrophe induction [171]. The fisetin-enhanced paclitaxel-induced mitotic catastrophe triggered cytoprotective autophagy, subsequently changing to autophagic cell death, which led to enhanced cytotoxicity [171].

Treatment with the ethyl acetate fraction of Glycyrrhiza uralensis extract that contains liquiritin, isoliquiritin, and isoliquirigenin decreased the viability of A549 cells, induced cell cycle arrest at G2/M phase, and apoptosis [172]. The ethyl acetate fraction significantly decreased the protein expression of PCNA, MDM2, phospho-GSK-3β, phospho-Akt, phospho-c-Raf, p-PTEN, caspase-3, pro-caspase-8, pro-caspase-9, PARP, and Bcl-2, and increased the expression of p53, p21, and Bax in a concentration-dependent manner, which suggested that the antitumor effects of liquiritin, isoliquiritin, and isoliquirigenin are orchestrated by the crosstalk among p53, Bcl-2 family, caspase cascades, and the Akt pathway [172].

Experimental evidence for the protective effects of several beverages and plant extracts against lung cancer through different mechanisms has been documented [15,173,174,175,177,178,179]. For example, the decoction extract of *Eucalyptus globulus* Labill. decreased the viability of H460 cells in a concentration-dependent manner, which was correlated with cell cycle arrest at the G0/G1 phase, decrease in cell proliferation, and increase in the expression of p53, p21, and cyclin D1 proteins [173]. Polyphenol compounds that were isolated from Selaginella tamariscina suppressed the migration of A549 cells by targeting matrix metalloproteinases (MMPs) [174]. The antitumor activity of the polyphenol-containing rosemary extract was associated with the inactivation of the Akt/mTOR signaling pathway [175]. The inhibitory effect of red wine on the proliferation and lonogenic survival of A549 cells was associated with the inhibition of basal and EGF-stimulated Akt and Erk phosphorylation and increased total and phosphorylated p53 levels [15]. Magnolol and polyphenol mixture derived from Magnolia officinali significantly suppressed the expression levels and function of class I histone deacetylases (HDACs) and enriched the histone acetyl mark (H3K27ac) in the promoter region of DR5, which is a key protein in the death receptor signaling pathway [177]. Pomegranate concentrate that was administered via drinking bottle to cigarette smoking (CS)-exposed mice prevented the formation of CS-induced lung nodules by reducing the mitotic activity and HIF-1αexpression in CS-exposed animals [178]. Oral administration of Achyranthes aspera (PCA) extract to urethane primed lung cancerous mice increased the expression and activities of antioxidant enzymes GST, GR, CAT, and SOD, decreased the expression and activity of LDH, downregulated the expression of pro-inflammatory cytokines IL-1β, IL-6, and TNF-α, along with TFs, NF-κB, and Stat3, and increased expression of Bax and p53 [179]. In addition, PCA was found to counteract urethane-mediated conformational changes of DNA evident by the shift in guanine and thymine bands in Fourier Trans-form Infrared (FTIR) spectroscopy, which suggests that the anticancer activity of PCA is associated with its immunomodulatory role and DNA conformation restoring effect [179].

## 5. Conclusions and Future Perspectives

Taken together, many years of research on the mechanisms of anticancer action of natural polyphenols have yielded an amazing amount of information. Strong lines of evidence have confirmed that certain natural polyphenols possess potential antitumor activities against lung cancer, which is the leading cause of cancer death in the United States and worldwide. Encouraging data from preclinical studies that were conducted in cell cultures and tumor models have provided much insight into a broad spectrum of molecular mechanisms underlying the anti-proliferative, anti-migratory, anti-metastasis, anti-angiogenic, and pro-apoptotic effects of various bioactive natural polyphenols in lung cancer. However, given that current chemotherapies for lung cancer have not advanced dramatically despite our increased knowledge base, much research is still needed to pave the way for the optimal integration of bioactive polyphenols with traditional chemotherapeutic regimens for lung cancer treatment, and for a full exploration of the polyphenol compounds that have the potential to form the basis for novel anticancer drugs of the future. In addition to continually refining and expanding our knowledge of the molecular mechanisms by which natural polyphenols exert their antiproliferative and proapoptotic activities against lung cancer, future research endeavors should also focus on the mechanistic understanding of bioavailability and the biodistribution process of natural polyphenols, which has been considered to be a challenging research field [224,225]. With the enhanced insight into the factors controlling tumor uptake of phenolic compounds and the advent of innovative drug delivery technologies, it is anticipated that new exploitable avenues will be opened for improved delivery of bioactive natural polyphenols to the site of action, thereby advancing their therapeutic utility in the treatment of lung cancer.

## Figures and Tables

**Figure 1 cancers-11-01565-f001:**
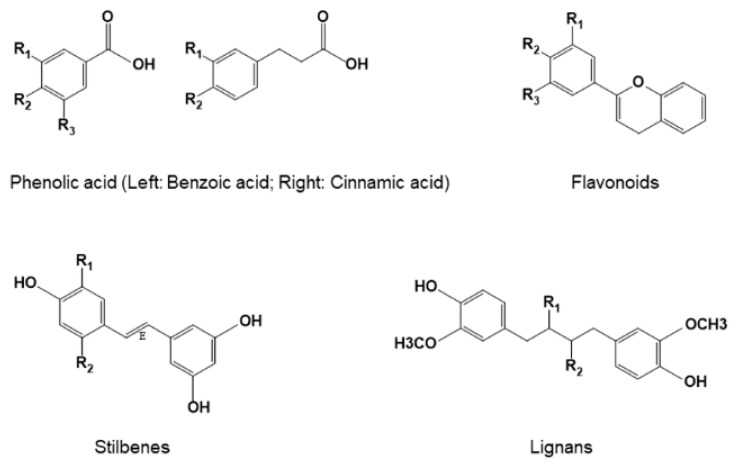
Chemical structures of different classes of natural polyphenols.

**Figure 2 cancers-11-01565-f002:**
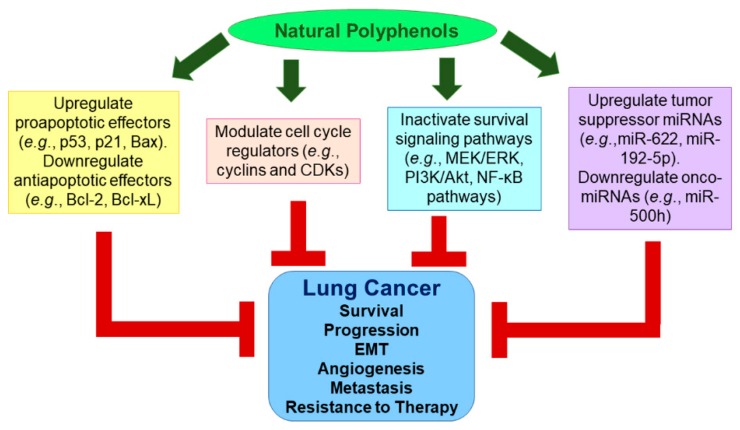
The role of bioactive natural polyphenols in lung cancer therapy.

**Figure 3 cancers-11-01565-f003:**
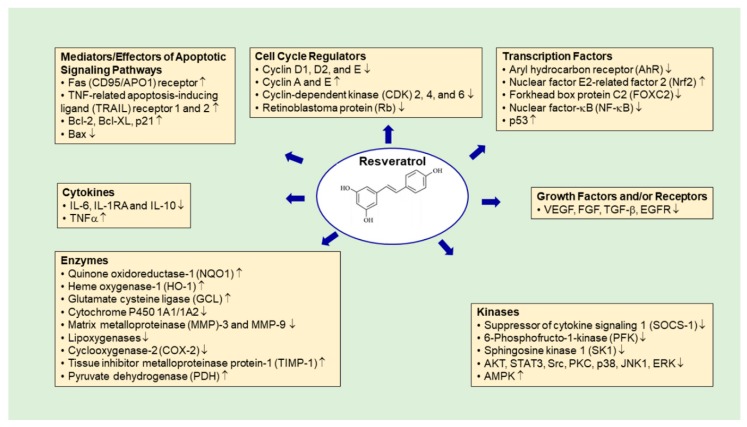
Molecular Mechanisms of Antitumor Activities of Resveratrol in Lung Cancer.

**Table 1 cancers-11-01565-t001:** Preclinical Studies on the Mechanisms Underlying the Antitumor Activities of Natural Polyphenols.

Polyphenol Compounds or Extracts	Mechanisms	In Vitro and/or In Vivo Models	References
Resveratrol	Induction of apoptosis by up-regulation of p53 and p21, activation of the caspases and disruption of the mitochondrial membrane complex. Cell cycle arrest at the G_1_ phase. Alterations in expressions of cyclin A, chk1, CDC27, and Eg5. Anti-tumor effect mediated by transforming growth factor-β pathway, particularly through the Smad proteins. i.e., down-regulation of the Smad activators 2 and 4 and up-regulation of the repressor Smad 7	A549 human NSCLC cell line	[90]
Resveratrol	Induction of apoptosis as a result of mitochondrial depolarization, release of cytochrome c from the mitochondrial compartment to the cytoplasm, apoptosis-inducing factor translocation from the mitochondrial compartment to the nucleus, and altered protein levels of Bcl-2, Bcl-xL and Bax	H446 human SCLC cells	[91]
Resveratrol	Induction of TRAIL-mediated apoptosis through suppression of NF-κB and downregulation of anti-apoptotic factors Bcl-2 and Bcl-xl	A549 and HCC15 human NSCLC cells	[92]
Resveratrol	Suppressed M2-like polarization of tumor associated macrophages and inhibited STAT3 activity	A549 and H1299 human NSCLC cells. Lewis lung cancer (LLC) s.c. xenograft model (Intraperitoneal (i.p.) administration) ^a^	[93]
Resveratrol	Resveratrol enhancing the effects of cisplatin on inhibition of cancer cell proliferation, induction of cell apoptosis, depolarization of mitochondrial membrane potential, release of cytochrome c, upregulation of Bax, downregulation of Bcl-2	H520 and H838 human NSCLC cell lines	[94]
Resveratrol	Upregulation of p21 and TRAIL receptor 1 and 2 expression, and downregulation of Bcl2, cyclin D, NF-κB and IKK1 expression	A549 human NSCLC cell line	[95]
Resveratrol	Increase in production of hydrogen peroxide (H_2_O_2_), activation of Bid, PARP and caspase 8, and downregulation of pEGFR, pAkt, c-FLIP and NF-κB protein expression	H460 human NSCLC cells	[96]
Resveratrol	Suppress of tumor cell growth via an apoptosis-independent mechanism involving induction of premature senescence by increasing P53 and p21 expression and ROS production and decreasing EF1A expression	A549 and H460 human NSCLC cell lines	[97]
Resveratrol	Enhancing ionizing radiation through increased production of ROS, and induction of DNA double-strand breaks and senescence induction	A549 and H460 human NSCLC cell lines	[98]
Resveratrol	Resveratrol overcoming gefitinib resistance by increasing the intracellular gefitinib concentration through inhibition of CYP1A1 and ABCG2 and by inducing cell apoptosis, autophagy, cell cycle arrest and senescence through increase in expression of cleaved caspase-3, LC3B-II, p53 and p21	PC9/G human NSCLC cells	[99]
Resveratrol	Resveratrol-enhanced erlotinib-mediated apoptosis through decreasing survivin expression and induction of PUMA expression	H460, A549, PC-9 and H1975 human NSCLC cell lines	[100]
Resveratrol	Resveratrol-enhanced etoposide-Induced cytotoxicity through down-regulating ERK1/2 and AKT-mediated X-ray repair cross-complement group 1 (XRCC1) protein expression	H1703 and H1975 human NSCLC cell lines	[101]
Resveratrol	Modulation of the expression of specific miRNAs with potential target genes involved in apoptosis, cell cycle regulation, cell proliferation, and differentiation	A549 human NSCLC cell line	[102]
Resveratrol	Upregulation of miR-622 leading to suppression of K-Ras mRNA translation without affecting its accumulation levels	16HBE-T human bronchial epithelial cell line and H460 human NSCLC cell line.	[103]
Resveratrol	Inhibition of lung cancer progression by downregulating miR-500h, which subsequently leads to downregulation of PP2A expression, inactivation of AKT/NF-kB and downregulation of FOXC2	CL1-5, A549, H322, H520 and H1435 human NSCLC cell lines	[104]
Resveratrol	Induction of G2/M cell cycle arrest through downregulation of checkpoint protein cyclin B1. Induction of apoptosis by increasing p53 and p21 expression and the release of cytochrome c in the cytosol	A549 human NSCLC cell line	[105]
Resveratrol	Inhibition of A549 cell proliferation through the reduction of the ratio of Bcl-2/Bax through activation of p53, thus activating the caspase-3- dependent apoptotic cascade and induces apoptosis	A549 human NSCLC cells.	[106]
Resveratrol	Attenuated A549 cell-induced platelet secretion and angiogenic responses in vitro and suppressed A549 lung cancer metastasis and angiogenesis in vivo through inhibition of platelets-mediated angiogenic responses induced by [106] adenosine diphosphate (ADP) through increased cGMP generation and cGMP-mediated vasodilator-stimulated phosphoprotein phosphorylation along with reduced intracellular Ca^2+^ mobilization	A549 human NSCLC cells, and A549 subcutaneous (s.c.) xenograft tumors in nude mice (i.p. administration) ^a^	[107]
Resveratrol	Anticancer effects attributable to inhibition of STAT-3 Signaling	A549 human NSCLC cells	[108]
Resveratrol	Inhibition of anchorage-dependent and -independent growth of NSCLC cells by decreasing EGFR and downstream kinases Akt and ERK1/2 activation, and subsequent impairment of hexokinase II (HK2)-mediated glycolysis by inhibiting HK2 expression mediated by the Akt signaling pathway	H460, H1650 and HCC827 human NSCLC cells. H460 s.c. xenograft model (i.p. administration) ^a^	[109]
Resveratrol	Synergism between Resveratrol and Metformin attributable to the suppression of DNA damage based on the downregulation of γH2AX/p53/p-chk2, inhibition of cell cycle progression via modulation of cyclin E/cdk2, Rb, p21 cyclin B1/cdk1 and plk1/cdc25c and enhancement of DNA repair indicated by the upregulation of p53R2	A549 human NSCLC cells	[110]
Resveratrol	Inhibition of the release of IL-6 and VEGF for co-cultured A549 lung cancer cells and adipose-derived mesenchymal stem cells	Co-cultured A549 human lung cancer cells and adipose-derived mesenchymal stem cells	[111]
Resveratrol	Induction of cell cycle arrest in the G0/G1 phase by downregulating the expression levels of cyclin D1, cyclin-dependent kinase (CDK)4 and CDK6, and upregulating the expression levels of the CDK inhibitors, p21 and p27	A549 human NSCLC cell line	[112]
Resveratrol	miR-200c sensitized tumor cell response to resveratrol by targeting reversion-inducing cysteine-rich protein with Kazal motifs (RECK), followed by activation of the JNK signaling pathway and ER stress	H460 human NSCLC cell line	[113]
Resveratrol	Suppression of invasion and metastasis through reversal of TGF-β1-induced EMT through increasing E-cadherin expression and repressing Fibronectin, Vimentin, Snail1 and Slug expression	A549 human NSCLC cell line	[114]
Resveratrol	Enhancing the radiosensitivity through NF-κB inhibition and S-phase arrest	NCI-H838 human NSCLC cell line	[115]
Resveratrol	Anti-metastasis effect attributable to the inhibition of expression of MMP-9/MMP-2 by suppression of HO-1, which in part results from the suppression of NF-κB-dependent signaling pathway	A549 human NSCLC cell line	[116]
Resveratrol	Inhibition of the proliferation of SPC-A-1/CDDP cells, induction of apoptosis and cell cycle arrest at phase between G0-G1 and S phase or at the G2/M phase by downregulating survivin	Human multidrug-resistant SPC-A-1/CDDP cells	[117]
Resveratrol	Anti-proliferative effect associated with inhibition of the phosphorylation of the retinoblastoma protein (pRB) and induction of cyclin-dependent kinase (Cdk) inhibitor p21WAF1/CIP. Induction of apoptosis associated with activation of caspase-3, shift in Bax/Bcl-xL ratio and inhibition of transcriptional activity of NF-κB	A549 human NSCLC cell line	[118]
Resveratrol loaded gelatin nanoparticles	Induction of cell death through inhibition of cell cycle progression and constitutive NF-κB activation by altering the expression of p53, p21, caspase-3, Bax, Bcl-2 and NF-κB	H460 human NSCLC cell line	[119]
SS28 (a synthetic Resveratrol analog)	Inhibition of Tubulin polymerization during cell division to cause cell cycle arrest at G2/M phase of the cell cycle	A549 human NSCLC cell line	[120]
4,4’-Dihydroxy-*trans*-stilbene (DHS) (a resveratrol analog)	Inhibition on anchorage-dependent or -independent cell growth, leading to impairment of the cell cycle progression with reduction of cell numbers arresting at the G1 phase	Murine Lewis lung carcinoma (LLC) cell line	[121]
Curcumin and resveratrol alone or in combination	Improvement of lung histoarchitecture and ultrahistoarchitecture during benzopyrene-induced lung carcinogenesis in mice	3,4-Benzopyrene-induced mouse lung carcinoma model (Oral (p.o.) administration) ^a^	[122]
Resveratrol and dibenzoylmethane	Induction of apoptosis through activation of caspase-9 and caspase-3 and subsequent cleavage of PARP	A549 and CH27 human NSCLC cell lines	[123]
Heyneanol A (HA) (A tetramer of resveratrol)	Induction of caspase-mediated cancer cell apoptosis by inducing cleavage of caspase-9 and caspase-3 and suppression of basic fibroblast growth factor (bFGF)-induced tumor angiogenesis.	In vivo Lewis lung tumor model (i.p. administration) ^a^	[124]
EGCG, ECG, EGC and EC	Induction of apoptosis through a p53-dependent pathway.	A549 human NSCLC cell line	[125]
EGCG	Induction of G2-M arrest. Incorporation into cytosol and nuclei	PC-9 human NSCLC cell line	[126]
EGCG	EGCG inhibited cell growth through decreasing the phosphorylation of Akt and ERK irrespective of EGFR-, ALK- or ROS1-dependency. The antiangiogenic effect of EGCG might be attributable to the inhibition of HIF-1α	PC-9, RPC-9, H1975, H2228 and HCC78 human NSCLC cell lines (EGFR- or fusion gene-driven tumor cells) and xenograft models (p.o. administration) ^a^	[127]
EGCG	Downregulation of gene expression of NF-κB inducing kinase (NIK), death-associated protein kinase 1 (DAPK 1), RhoB and tyrosine-protein kinase (SKY), and upregulation of the retinoic acid receptor alpha1 gene expression	PC-9 human NSCLC cell line	[128]
EGCG	Induction of miRNA profile changes, which modulate several regulatory networks associated to AKT, NF-κB, MAP kinases, and cell cycle	4-(Methylnitrosamino)-1-(3-pyridyl)-1-butanone (NNK) induced mouse lung cancer (p.o. administration) ^a^	[129]
EGCG	Co-treatment with celecoxib synergistically inducing apoptosis by upregulation of growth arrest and DNA damage-inducible 153 (GADD153) through the ERK signaling pathway	A549, ChaGo K-1 and PC-9 human NSCLC cell lines	[130]
EGCG	Co-treatment of EGCG with cisplatin resulting in proliferation inhibition, cell cycle arrest in G1 phase, increase in apoptosis along with inhibition of DNA methyltransferase (DNMT) activity and histone deacetylase (HDAC) activity, reversal of hypermethylated status and downregulated expression of *GAS1*, *TIMP4*, *ICAM1* and *WISP2* genes	Cisplatin-resistant A549 (A549/DDP) human NSCLC cell line and A549/DDP xenograft tumor model (i.p. administration)	[131]
EGCG and Luteolin	Enhanced antitumor effect attributable to ATM (ataxia telangiectasia mutated) kinase-dependent Ser15 phosphorylation of p53 as a consequence of DNA double strand break	H292, A549 and H460 human NSCLC cell lines expressing wild-type p53; A549 xenograft tumor model (p.o. administration) ^a^	[132]
Tea polyphenols	Upregulation of p53 expression and downregulation of Bcl-2 expression with no influence on H-Ras and c-Myc expressions	NCI-H460 human NSCLC cell line	[133]
Black tea polyphenols	Inhibition of Cox-1 and induction of caspase-3 and caspase-7 expression	3,4-Benzopyrene induced mouse lung tumor model	[134]
Black tea polyphenols	Suppressing cell proliferation and inducing apoptosis	3,4-Benzopyrene induced mouse lung tumor model	[135]
Green tea polyphenols	Preventive effect against lung cancer by upregulating p53 and downregulating Bcl-2	3,4-Benzopyrene induced rat lung tumor model	[136]
Tea polyphenols	Increase in p53 expression and decrease in Bcl-2 expression	3,4-Benzopyrene-induced rat lung carcinoma model	[137]
Tea polyphenols	Inhibition of Akt and cyclooxygenase-2 expression, and inactivation of nuclear factor-kappa B via blocking phosphorylation and subsequent degradation of IkappaB alpha	Diethylnitrosoamine- induced mouse lung tumor model (p.o. administration) ^a^	[138]
Green tea polyphenols	TAM67-mediated changes in gene expression involving the downregulation of activator protein-1 (AP-1)	H1299 human NSCLC cell line and SPON 10 mouse lung tumor cell line	[139]
Green tea extracts	Modulation of the expression of 14 proteins involved in calcium-binding, cytoskeleton and motility, metabolism, detoxification, or gene regulation	A549 human NSCLC cell line	[140]
Green tea polyphenols	Synergistic antitumor effect with atorvastatin attributable to increased apoptosis, reduced Mcl-1 level and increased cleaved caspase-3 and cleaved poly(ADP)-ribose polymerase (PARP)	H1299 and H460 human NSCLC cell lines. 4-(Methylnitrosaminao)-1-(3-pyridyl)-1-butanone induced mouse lung tumor model (p.o. administration) ^a^	[141]
Green tea extract	Induction of protective autophagy	A549 human NSCLC cell line	[142]
Thymoquinone (TQ)	Upregulation of Bax and downregulation of Bcl-2 expression and increase in the Bax/Bcl-2 ratio. Decrease in the expression of cyclin D, NF-κB and IKK1 and increase in the expression of p21 and TRAIL receptor 1 and 2 expression	A549 human NSCLC cell line	[95]
Curcumin	Induction of apoptosis through p53-independent pathway by downregulation of Bcl-2 and Bcl-xL expression	A549 and H1299 human NSCLC cell lines	[143]
Curcumin	Induction of cell cycle arrest at the G1/S phase and apoptosis through up-regulation of *GADD45* and *GADD153*	PC-9 human NSCLC cell line	[144]
Curcumin	Induction of cell cycle arrest at the G2/M phase and apoptosis through upregulation of Bax and Bad expression, downregulation of Bcl-2, Bcl-xL and XIAP expression, increase in ROS, intracellular Ca^2+^ and endoplasmic reticulum stress, activation of GRP78 and GADD153 proteins and FAS/caspase-8 pathway	H460 human NSCLC cell line	[145]
Curcumin	Induction of apoptosis through a mitochondria-dependent mechanism as manifested by the decrease in the mitochondrial membrane potential, releasing cytochrome c from mitochondria to cytoplasm	A549 human NSCLC cell line	[146]
Curcumin	Induction of apoptosis via the ROS-mediated mitochondrial pathway accompanied by increased Bax expression and decreased expression of Bcl-2 and Bcl-xL	H446 human SCLC cell line	[147]
Curcumin	Inhibition of tumor cell proliferation and induction of apoptosis through upregulation of miR-192-5p and suppression of the PI3K/Akt signaling pathway	A549 human NSCLC cell line	[148]
Curcumin	Enhancing autophagy and apoptosis through inaction of PI3K/mTOR signaling pathway	A549 and H1299 human NSCLC cell lines	[149]
Curcumin	Induction of autophagy leading to suppression of tumor cell proliferation.	A549 human NSCLC cell line	[150]
Curcumin	Inhibition of tumor cell invasion and metastasis through attenuating GLUT1/MT1-MMP/MMP2 pathway	A549 human NSCLC cell line and xenograft tumor model (i.p. administration) ^a^	[151]
Curcumin	Inhibition of tumor cell metastasis through inhibition of the adiponectin/NF-κB/MMPs signaling pathway	A549 human NSCLC cell line and xenograft tumor model (i.p. administration) ^a^	[152]
A501 (Curcumin analogue)	Induction of cell cycle arrest at the G2/M phase and apoptosis through decreasing the expression of cyclinB1, cdc-2, Bcl-2, while increasing the expression of p53, cleaved caspase-3 and Bax	A549 and H460 human NSCLC cell lines	[153]
Curcumin and gefitinib	Potentiating the antitumor effect of gefitinib in gefitinib-resistant tumor cells through induction of endogenous EGFR protein degradation and downregulation of EGFR and AKT protein expression. Reduction of the gefitinib-induced villi damage and apoptosis in mouse intestine through attenuating gefitinib-induced p38 activation	CL1-5, A549 and H1975 human NSCLC cell lines and xenograft models (p.o. administration) ^a^	[154]
Curcumin and carboplatin	Synergistic antitumor activity mediated by multiple mechanisms involving suppression of NF-kB via inhibition of the Akt/IKKα pathway, enhancement of ERK1/2 activity and downregulation of MMP-2 and MMP-9 expression	A549 human NSCLC cell line	[155]
Quercetin	Induction of cell cycle arrest at the G2/M phase and apoptosis through increased expression of cyclin B1 and phosph-cdc2 (T161), survivin, total p53, phosphor-p53 (S15) and p21 proteins	A549 and H1299 human NSCLC cell lines	[156]
Quercetin	Induction of apoptosis through activation of MEK-ERK pathway, inactivation of Akt and alteration in the expression of Bcl-2 family	A549 human NSCLC cell line	[157]
Quercetin	Proapoptosis activity through multiple mechanisms including upregulated the expression of genes associated with the death pathway, the JNK pathway, the IL1 receptor pathway, the caspase cascade, the NF-*κ*B pathway and cell cycle arrest, and downregulated the expression of genes related to cell proliferation	H460 human NSCLC cell line	[158]
Quercetin	Anti-invasion activity through inhibition of monocarboxylate transporter 1	A110L human lung cancer cell line	[159]
Quercetin	Targeting aurora B kinase	A549 human NSCLC cell line and xenograft model (i.p. administration) ^a^	[160]
Quercetin	Trigger Bcl-2/Bax-mediated apoptosis, necrosis and mitotic catastrophe. Inhibition of cell migration through disassembly of microfilaments, microtubules and vimentin filaments and inhibition of vimentin and N-cadherin expression	A549 human NSCLC cell line	[161]
Quercetin	Suppression of in vitro cell migration/invasion and in vivo bone metastasis through inhibition of Snail-dependent Akt activation and Snail-independent ADAM9 expression pathways	A549 and HCC827 human NSCLC cell lines and A549 xenograft model (i.p. administration) ^a^	[162]
Quercetin and chrysin	Suppressed the secretion of cytokines, IL-1β, IL-6, TNF-α and IL-10, and decreased the phosphorylation of IKKβ and IκB, the nuclear level of p65 (NF-κB) as well as the expression of MMP-9 in A549cells exposed to nickel	A549 human NSCLC cell line	[163]
Quercetin and trichostatin A	Enhancing the antitumor activity of trichostatin A through upregulation of p53 expression	A549 and H1299 human NSCLC cell lines and A549 xenograft model (i.p. administration) ^a^	[164]
Quercetin and gemcitabine	Promoting apoptosis and sensitizing tumor response to gemcitabine via inhibition of HSP70 expression	A549 and H4650 human NSCLC cell lines	[165]
Caffeic acid phenethyl ester (CAPE)	Upregulation of Bax, p21 and TRAIL receptor 1 and 2 expression, and downregulation of cyclin D expression	A549 human NSCLC cell line	[95]
Pterostilbene	Exhibition of p53-dependent chemotherapeutic effects through the ATM/CHK/p53 tumor suppressive pathway leading to cell senescence	Precancerous human bronchial epithelial cell lines, HBECR and HBECR/p53i, with normal p53 and suppressed p53 expression, respectively	[166]
Bakuchiol	Increase in reactive oxygen species production, decrease in mitochondrial membrane potential (∆Ψm), cell cycle arrest at S phase, caspase 9/3 activation, p53 and Bax up-regulation, and Bcl-2 downregulation	A549 human NSCLC cell line	[167]
Chlorogenic acid (CGA)	Decrease in hypoxia-induced HIF-1α protein level and suppression of the transcriptional activity of HIF-1α under hypoxic conditions, leading to antiangiogenic activity through inhibition of HIF-1α/AKT pathway and decrease in VEGF expression	A549 human NSCLC cell line	[168]
Fisetin	Inhibition of cell growth through concomitant suppression of PI3K/Akt and mTOR signaling	A549 and H1792 human NSCLC cell lines	[169]
Fisetin	Enhancing cisplatin cytotoxicity in cisplatin-resistant cells by modulation of the MAPK/survivin/caspase pathway	A549 human NSCLC cell line	[170]
Fisetin	Synergistic interaction between paclitaxel and fisetin due to the induction of mitotic catastrophe probably through the promotion of multipolar spindle formation. Mitotic catastrophe induced protective autophagy against apoptosis, which then switched to the autophagic cell death	A549 human NSCLC cell line	[171]
Liquiritin, isoliquiritin and isoliquirigenin	Induction of apoptosis and cell cycle at the G2/M phase by increasing p53, p21 and BAX expression and decreasing PCNA, MDM2, p-GSK-3β, p-Akt, p-c-Raf, p-PTEN, caspase-3, pro-caspase-8, pro-caspase-9, PARP and Bcl-2 expression	A549 human NSCLC cell line	[172]
Eucalyptus globulus Labill	Cell cycle arrest in the G0/G1 phase. Increase in the expression of p53, p21 and cyclin D1 proteins	NCI-H460 human NSCLC cell line	[173]
Polyphenols isolated from	Suppressed cell migration by targeting MMP-9. Induced cell apoptosis through intrinsic apoptosis pathways, accompanied by increasing the expression of Bax and caspase-3	A549 human NSCLC cell line	[174]
Rosemary extract	Reduced total and phosphorylated/activated Akt, mTOR and p70S6K levels	A549 human NSCLC cell line	[175]
Salvianolic acid A	Salvianolic acid A enhanced sensitivity to cisplatin through suppression of the c-met/AKT/mTOR signaling pathway	A549 human lung cancer cisplatin resistance cell line (A549/DDP)	[176]
Red wine	Inhibition of basal and EGF-stimulated Akt and Erk signals and enhancement of total and phosphorylated levels of p53, leading to inhibition of A549 cell proliferation and clonogenic survival	A549 human NSCLC cell line	[15]
Magnolol and polyphenol mixture (PM) derived from Magnolia officinalis	Induction of cell apoptosis by arresting the cell cycle in the G0/G1 phase while simultaneously activating various pro-apoptotic signals, including TRAIL-R2 (DR5), Bax, caspase 3, cleaved caspase 3, and cleaved PARP	A549 and H1299 human NSCLC cell lines	[177]
Pomegranate juice	Attenuated the formation of lung nodules and reduced PHH3 (marker of mitotic activity) and HIF-1*α* expression	Two-month-old adult male AJ mice (p.o. administration) ^a^	[178]
Polyphenolic compounds of Achyranthes aspera (PCA) extract	Downregulation of the expression of pro-inflammatory cytokines IL-1β, IL-6 and TNF-α, TFs, NF-κB and Stat3, and upregulation of the expression of pro-apoptotic proteins Bax and p53. Increase in the activities and expression of antioxidant enzymes GST, GR, CAT, and SOD. Decrease in the activity and expression of LDH enzymes	Urethane-induced mouse lung cancer in vivo model (p.o. administration) ^a^	[179]
Bilberry extract (BE), genistein (GEN), delphinidin-3-*O*-glucoside (D3G), delphinidin (DE), gallic acid (GA), and phloroglucinol aldehyde (PGA)	Antagonistic interactions of BE, D3G, DEL and GEN with erlotinib. No effect of GA on erlotinib, while synergistic interaction of PGA with erlotinib. Mechanism unknown	A431 human epithelial cell line	[180]

Note: a. The route of administration for individual natural polyphenols evaluated in in vivo studies.

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
