# Peer review of "Molecular Insights into Potential Contributions of Natural Polyphenols to Lung Cancer Treatment"

_cancers, 2019, doi:10.3390/cancers11101565_

Round 1

Reviewer 1 Report

The manuscript is well written and it requires minor revision.

1. As author focus on natural polyphenols. Two main polyphenols is missing in this review.

2. Curcumin and its other form should include.

3. Quercetin is yet another major compound and it need to include in detail

4. Molecular mechanism of the polyphenol role in the prevention or treatment should contain some figures.

5. The author should also discuss about the bioavailability in the conclusion and the author suggestion for the enhanced bioavailability of these compounds

Author Response

Reply to Point 1 - 3: We appreciate the constructive suggestions from Reviewer 1. Two sections have been added to the manuscript to review the molecular mechanisms associated with the antitumor activities of curcumin and quercetin in lung cancer.

Reply to Point 4: Figure 2 that depicts the role of natural polyphenols in lung cancer therapy has been added to the manuscript.

Reply to Point 5: A couple of sentences have been added to the end of the “Conclusions and Future Perspectives” section, in which we touched on the future prospect of the area of intensive research concerned with the bioavailability and targeted delivery of phenolic compounds for the treatment of lung cancer.

Reviewer 2 Report

This review is very well-written and articulated. It encompasses a thorough revision of the literature dealing with the original works, most of them showing the molecular mechanisms responsible for the antitumor potential of natural polyphenols with focus on lung cancer. A comprehensive table and one figure illustrate the text, which entails almost 200 references.

This review deals with an area of research (polyphenols for cancer prevention or treatment) that is of upmost interest for many researchers and clinicians, so in my opinion it could facilitate the apprehension of the great amount of information that is emerging from both in vitro and in vivo studies using natural polyphenols due to their anticancer potential.

I do not have any further comments to the review, and I think it could be disseminated in its actual form.

Author Response

Reply: We appreciate the encouraging comments from Reviewers 2.

Reviewer 3 Report

This review compelled evidence from in vitro and in vivo studies demonstrating the ability of natural polyphenols to suppress lung cancer and discussed the possible molecular mechanisms. This article is interesting, well organized and comprehensive described.

There are a few concerns.

In the section 3, the authors mentioned about that Polyphenols can be classified into four main groups, including phenolic acids, flavonoids, stilbenes and lignans. It is better to provide the basic structures of these four groups of compounds. That will help the readers to understand the following text. Because polyphenols given orally rather than intraperitoneally are converted to various metabolites rapidly in vivo, the routes by which the polyphenols are given in animal studies affect their efficiencies and underlying mechanisms. Thus, the routes used in the animal studies should be described. That may partly explain why the mechanisms by which those polyphenols exert their anticancer effects in vivo are different from those found in vitro. As mentioned by the authors in the conclusion”… current chemotherapies for lung cancer have not advanced dramatically despite our increased knowledge base, much research is still needed to pave the way for the optimal integration of bioactive polyphenols with traditional chemotherapeutic regimens for lung cancer treatment….”. In vitro study, the parent compounds of polyphenols are used to perform studies and showed various physiological activities including anticancer effects. However, as mentioned above, in vivo studies demonstrated there are different efficiency between these compounds given by oral or by intraperitoneal injection. In most cases, phytochemicals given orally possess less biological efficiency than the same compound given intraperitoneally because of metabolic conversion (Biomed Res Int. 2014;2014:580626. doi: 10.1155/2014/580626. Oral and intraperitoneal administration of quercetin decreased lymphocyte DNA damage and plasma lipid peroxidation induced by TSA in vivo). However, human takes these compounds by oral intake traditionally and in clinical trials. Regarding as a component of chemotherapy, it may be necessary to consider giving these polyphenols by i.v. injection or developing new approach to decrease the metabolic conversion. The authors are suggested to discuss about this issues. page16, IL-1b should be IL-1β.

Author Response

Reply: (1) Figure 1 that shows the chemical structures of different classes of natural polyphenols has been added to the manuscript.

(2) We appreciate the insightful comments from Reviewer 3. The information on the route of administration for individual natural polyphenols evaluated in vivo has been added to Table 1. It was shown that both oral and intraperitoneal administration were commonly used in animal studies. Results of those studies indicated that the in vivo tumor responses to polyphenol treatment were mostly comparable with those observed in vitro in terms of the molecular mechanisms involved in the antitumor effects of polyphenol compounds. Since the primary focus of this review is on the molecular mechanisms underlying the antitumor activities of natural polyphenols in lung cancer, we did not discuss the influence of various administration routes on the antitumor efficacy of polyphenol compounds. Nonetheless, given the complexity of a myriad of conditions that can impact the distribution of polyphenol compounds in lung tumors, we have included a couple of sentences at the end of the “Conclusions and Future Perspectives” section to touch on the future prospect of the area of intensive research concerned with the bioavailability and targeted delivery of phenolic compounds for lung cancer therapy.

(3) IL-1b has been changed to IL-1β.